# A Comparison of the Fifth World Health Organization and the International Consensus Classifications of Mature T-Cell Lymphomas

**DOI:** 10.3390/ijms241814170

**Published:** 2023-09-16

**Authors:** Pier Paolo Piccaluga, Shaimaa S. Khattab

**Affiliations:** 1Biobank of Research, IRCCS Azienda Opedaliera-Universitaria di Bologna, 40138 Bologna, Italy; 2Department of Medical and Surgical Sciences, Bologna University School of Medicine, 40138 Bologna, Italy; 3Medical Research Institute, Hematology Department, Alexandria University, Alexandria 5310002, Egypt; shaymaa.khattab@alexu.edu

**Keywords:** World Health Organization classification, International Consensus Classification, T-cell lymphoma, T-follicular helper, gene expression profile, immunophenotype, prognosis, diagnosis, targeted therapy, personalized medicine

## Abstract

Peripheral T-cell lymphomas (PTCLs) are a rare subset of non-Hodgkin lymphomas that often carry significant difficulty in diagnosis and classification because of their rarity and biological complexity. Previous editions of the World Health Organization (WHO) classifications of hemopoietic neoplasms in 2001, 2008, and 2017 aimed to standardize hemopoietic neoplasm diagnosis in general. Since then, crucial clinico-pathological, immunophenotypic, and recent molecular discoveries have been made in the field of lymphomas, contributing to refining diagnostic criteria of several diseases, upgrading entities previously defined as provisional, and identifying new entities. In 2022, two different models were proposed to classify hematolymphoid neoplasms: the 5th edition of the WHO classification (WHO-HAEM5) and the International Consensus Classification (ICC). Of note, a common nosography is mandatory to ensure progress in health science and ensure the basis for a real precision medicine. In this article, the authors summarized the main differences with the previous fourth WHO edition and reviewed the main discrepancies between the two newest classifications, as far as PTCLs are concerned.

## 1. Introduction

For many years, the classification of hematolymphoid neoplasms has been heterogenous, particularly in different geographic settings (e.g., Europe vs. USA) and has basically relied on tissue architecture and cytologic appearance [1]. The advent of monoclonal antibodies in the late 1970s, and of molecular technologies later, led to an immunologic and genetic revolution, leading to new, more refined, and homogenous classifications in the 1990s [1].

The previous editions of the World Health Organization (WHO) classifications of hemopoietic (lymphoid and myeloid) neoplasms largely contributed to standardizing tumor classification worldwide, leading to more correct diagnoses, more refined prognostication, and, eventually, to possible individualized treatment for a fraction of patients affected by hematopoietic neoplasm.

In 2022, two different working groups updated the classification, namely the International Consensus Classification (ICC) and the WHO-HAEM-5, the fifth edition of the WHO classification, which now represent the two currently available options for both lymphoid and myeloid neoplasms [2,3,4,5].

Both the ICC and the WHO-HAEM5 classify lymphoid malignancies first according to the normal counterpart, as defined by the immunophenotype (i.e., B-cell lineage versus T- and NK-cell-associated neoplasms); these categories are further divided into precursor neoplasms versus mature neoplasms. In comparison to the previous WHO-HAEM4R, the broad categories of lymphoid malignancies in ICC and WHO-HAEM5 have not undergone any substantial modifications. However, changes to the nomenclature and categorization of some lymphoid neoplasms are included in both schemes.

In general, the WHO-HAEM5 groups mature T-cell malignancies into nine categories: mature T-cell and NK-cell leukemias, primary cutaneous T-cell lymphomas, intestinal T-cell and NK-cell lymphoid proliferations and lymphomas, nodal T-follicular helper (TFH) cell lymphoma, hepatosplenic T-cell lymphoma, anaplastic large-cell lymphoma, EBV-positive NK/T-cell lymphomas, EBV-positive T- and NK-cell lymphoid proliferations and lymphomas of childhood, and other peripheral T-cell lymphomas. By contrast, the ICC considered all entities as one single group of mature T-cell and NK-cell neoplasms (Table 1). There is no official information about how the two classifications are currently applied worldwide, and this certainly needed further investigation.

In this article, the authors reviewed and compared for the first time the two classification proposals as far as mature T-cell neoplasms are concerned, highlighting the most significant differences and discrepancies, aiming to offer a practical guide to pathological diagnosis [3,4,5,6,7]. To this aim, the authors compared the two official publications of the WHO and ICC, further referring to original research on the single entities as within the two classifications.

## 2. Classification of Mature T-Cell Neoplasms

Peripheral T-cell lymphomas (PTCLs) are a rare subset of non-Hodgkin lymphomas (NHL) that arise from mature (i.e., post-thymic or “peripheral”) T lymphocytes and NK cells [8,9]. With a few very rare exceptions, they are often aggressive tumors with a severe prognosis [7]. Asia has a higher prevalence of PTCLs than Europe and USA [8]. Reports about PTCLs in Africa are, conversely, scant, probably reflecting the common lack of immunohistochemistry, which is mandatory for their proper recognition. This regional variation is brought about by postulated genetic differences and different patterns of viral exposures linked to distinct PTCL entities, such as Epstein–Barr virus (EBV) and human T-lymphotropic virus 1 (HTLV-1) [10]. Furthermore, the varied prevalence of celiac disease also affects the distribution of some intestinal forms (see below).

As mentioned for NHL in general, T-cell malignancies are also sub-classified based on the normal counterparts and the immune-histological patterns; however, the clinical features play a major role, defined as nodal, extranodal, and leukemic forms (Table 1).

In the following, the different entities are discussed and the differences between ICC and WHO-HAEM5 are summarized.

## 3. Mature T-Cell Leukemias

### 3.1. T-Prolymphocytic Leukemia

The diagnostic criteria have been made uniform, based on the presence of post-thymic T lymphocytosis (>5 × 10^9^/L), and T prolymphocytes have the typical morphological features of prolymphocytes, like a moderate cell size with an irregular nuclear membrane, single round prominent nucleoli, and a basophilic cytoplasm with blebs. Inversion of chromosome 14 [inv(14)] or t(14;14) through conventional karyotyping and/or FISH is the most common cytogentic abnormality found in almost 77% of patients with T-PLL [11]. The T-cell phenotype is negative for TdT and CD1A, and positive for CD2, CD3, CD5, and CD7; the membrane expression of CD3 may be weak. High levels of CD52 expression, which represents a suitable therapeutic target [12,13], and a possible co-expression of CD4 and CD8 are reported [11,14].

Overexpression of the oncoprotein TCL1 can be shown via immunohistochemistry or flow cytometry, as the occurrence of the oncogeneic process is initiated by rearrangements of *TCL1A*/*B* or *MTCP1* [15].

### 3.2. T-Cell Large Granular Lymphocyte (LGL) Leukemia

T-cell large granular lymphocyte (LGL) leukemia is defined by both WHO-HAEM5 and ICC as a disorder of clonally expanded T-cell large granular lymphocytes that invade the bone marrow, spleen, and liver [16,17]. It was called T-cell large granular lymphocytic leukemia in WHO-HAEM4R, and as such remained in ICC [7]. The peripheral blood smear is characterized by the presence of mature lymphoid neoplastic cells that are larger than most circulating lymphocytes and have distinctive azurophilic granules containing acid hydrolases. The LGL leukemia immunophenotype includes CD3+, CD57+, and CD56-negative T cells, more often CD8+, corresponding to in vivo antigen-activated effector memory cytotoxic T cells [18,19].

*STAT3* seems to be relevant in the pathogenesis of the disease, being associated with neutropenia and shorter survival [16,17,20].

### 3.3. NK-Cell Large Granular Lymphocyte Leukemia

In the WHO-HAEM5, the term NK-LGL leukemia replaced the chronic lymphoproliferative disorder of NK cells. This nomenclature reflects the monoclonal or oligoclonal NK cells that differ from the neoplastic cells of T-LGL leukemia. Neutropenia is not as severe as in T-cell LGL leukemia. The predominant phenotype of NK cells is CD2+/CD3−/CD4−/CD8−/CD16+/CD56+, and the median absolute number of NK cells is 2.3 × 10^9^/L (2300/microL) [21,22]. Typically, CD57 is only weakly expressed [23,24]. The previous nomenclature, by contrast, was maintained by the ICC.

### 3.4. Aggressive NK-Cell Leukemia

The diagnostic standards for aggressive NK-cell leukemia have not changed in both WHO-HAEM5 and ICC compared to WHO-HAEM4R. When a patient has persistent peripheral blood lymphocytosis, aggressive clinical illness, neutropenia, anemia, coagulopathy, hemophagocytic syndrome, and LGLs are detected on a peripheral blood smear, aggressive NK-cell leukemia is suspected. The vast majority of aggressive NK-cell leukemias exhibit a phenotype that includes CD3−, CD4−, CD8+, CD16+, and CD56+; CD57 may be expressed in some cases [8,25]. Additionally, sequencing analysis enables the involvement of many mutations in particular disease-related pathways, such as *JAK*/*STAT* and *RAS*/*MAPK*, epigenetic modifiers (*KMT2D*, *TET2*, *CREBBP*), and immune checkpoints (*PDL1*, *PDL2*) [26]. *TP53* mutation has also been identified in many studies [27,28].

### 3.5. Adult T-Cell Leukemia/Lymphoma

The peripheral T-cell malignancy that arises from CD4-positive T cells that are latently infected with the human T-cell leukemia virus (HTLV) type 1 is still referred to as adult T-cell leukemia/lymphoma (ATLL) by both ICC and WHO-HAEM5. Generalized bulky lymphadenopathy, hepatosplenomegaly, immunosuppression, hypercalcemia, lytic bone lesions, CNS, and skin lesions are common manifestations of ATLL [29].

In both classifications, it is referred as a tumor more common in Asia and the Caribbean areas, where HTLV1 is endemic; however, it should be noted that HTLV1 has been found to be endemic in Eastern Europe too, namely in Romania. It is definitely necessary that an epidemiological study of ATLL is performed in Romania soon [30].

### 3.6. Sezary Syndrome

Sezary syndrome (SS) is characterized by erythroderma, generalized lymphadenopathy, and the presence of clonally linked neoplastic T cells with cerebriform nuclei (Sezary cells) in the skin, lymph nodes, and peripheral blood are the three symptoms that make up Sezary syndrome (SS). Despite being somehow related, SS is currently distinct from mycosis fungoides. In the WHO-HAEM5 classification, mycosis fungoides and Sézary syndrome are put into different groups, while in the ICC they are listed alongside each other.

A CD4:CD8 ratio of 10 or more, an enlarged CD4+ T-cell population, CD4^+^/CD7^−^ cells ≥ 40% or CD4^+^/CD26^−^ cells ≥ 30% [31], as well as the deletion of one or more T-cell antigens are also necessary requirements [32].

## 4. Primary Cutaneous T-Cell Lymphomas

Within the mature T/NK-cell neoplasms in WHO-HAEM5, primary cutaneous T-cell lymphoid proliferations and lymphomas (CTCL) make up a separate group and encompass nine different entities (Table 1). In the ICC, the sole observed change is the nomenclature of the primary cutaneous acral CD8+ T-cell lymphoma, as it is now classified as a primary cutaneous acral CD8+ T-cell LPD [3].

### 4.1. Mycosis Fungoides

Mycosis fungoides (MF) is the most common cutaneous T-cell lymphoma [33]. According to the WHO-HAEM5, it is distinct from SS even if they are similar and somehow related neoplasms; in fact, they are treated differently due to the different clinical behavior and cell of origin/cellular counterparts. For these reasons, SS is not even included among CTCLs, to highlight its primary location in peripheral blood and visceral organs and its aggressive clinical presentation and to allow for consideration in the differential diagnosis of mature T-cell leukemias (see above).

Despite MF being a single entity, within the folliculotropic category, two different patterns are recognized, namely clinical early vs. advanced stage, to account for their differing clinical outcomes. The other subgroups of MF are still recognized as subtypes (pagetoid reticulosis and granulomatous slack skin), as in the WHO-HAEM4 [7,31]. Regarding ICC, no significant changes regarding MF classification.

### 4.2. Primary Cutaneous Acral CD8+ T-Cell Lymphoproliferative Disease

Because of its highly indolent course and the fact that it just requires local therapies and surveillance, primary cutaneous acral CD8+ T-cell lymphoproliferative lymphoma is currently regarded as a lymphoproliferative illness rather than an overt lymphoma by both ICC and WHO-HAEM5. Even though 20% of patients do experience a local or more severe recurrence, only one patient had extracutaneous spread, according to current reports [34], and regardless of the kind of treatment, a 100% survival rate was recorded [35,36,37,38].

### 4.3. Primary Cutaneous Peripheral T-Cell Lymphoma, Not Otherwise Specified (NOS)

Primary CTCL-NOS is a totally new entry in the WHO-HAEM5. It is not included in the new ICC classification, though. Given that there is morphologic and immunophenotypic overlap across the many primary CTCL forms, a thorough medical history, appropriate clinical examination, and extensive pathologic and immunophenotypic study should all be performed before making a valid classification and diagnosis. If a specific entity cannot be diagnosed, the term primary CTCL-NOS has to be used [3,4,31,39].

## 5. Intestinal T-Cell and NK-Cell Lymphoid Proliferations and Lymphomas

### 5.1. Indolent NK-Cell Lymphoproliferative Disease of the Gastrointestinal Tract

Indolent NK-cell lymphoproliferative disease of the gastrointestinal tract (iNKLPD) is a new entity added to both the WHO-5 and the ICC. It can be asymptomatic or manifest with generalized gastro-intestinal (GI) symptoms. Usually, it spontaneously regresses, but occasionally new lesions develop [40,41]. The stomach and small and large intestines are typically affected. It has not been related to either EBV infection or celiac disease. It has a typical medium-sized cell with pale cytoplasm, eosinophilic granules, minimal epitheliotropism, and superficial invasion, and no necrosis from a morphological perspective. The phenotype is related to NK cells rather than T cells [40,42,43].

### 5.2. Indolent T-Cell Lymphoma of the Gastrointestinal Tract

Both the WHO-HAEM5 and the ICC revised the name of indolent T-cell lymphoproliferative disorder of the gastrointestinal tract, now included as indolent T-cell lymphoma of the gastrointestinal tract. The disease most commonly involves the small bowel or colon. Non-specific abdominal symptoms can be either chronic/persistent or recurrent. It is not related to celiac disease or EBV infection. At morphology, the invasion is superficial and constituted of small lymphoid elements with some degree of nuclear atypia but without epitheliotropism or focal necrosis. The molecular pathogenesis is largely related to *JAK2::STAT3* fusion, mutations of JAK-STAT pathway genes, and epigenetic modifier genes [44].

The phenotype of indolent T-cell lymphoma of the gastrointestinal tract is either CD4+, CD4+/CD8+, or CD4−/CD8−, with CD4+ cases sometimes displaying *STAT3::JAK2* fusions. In contrast, some CD8+ cases have been shown to harbor structural alterations involving the IL2 gene [45,46].

Indolent NK-cell lymphoproliferative disorder of the gastrointestinal tract is a new entity that has been added to both the WHO-HAEM5 and the ICC.

The other three entities among intestinal T-cell lymphomas, namely enteropathy-associated T-cell lymphoma, monomorphic epitheliotropic intestinal T-cell lymphoma, and intestinal T-cell lymphoma NOS, were not modified in either one of the two new classifications and remained as in WHO-HEM4.

## 6. Hepatosplenic T-Cell Lymphoma

Hepatosplenic T-cell lymphoma (HSTL) is an extremely aggressive form of extranodal lymphoma, distinguished by a hepatosplenic appearance without lymphadenopathy and a poor prognosis.

The neoplastic cells are typically CD3+, CD56+/−, CD4−, and CD5−/+. While the T-cell receptor gamma/delta (TCRG/D) is expressed in the majority of cases (75%), the remaining minority of cases are either TCRA/B+ (~20–25%) or, rarely, TCR-silent (<5%) [47,48,49,50].

Neither the ICC nor the WHO changed the descriptions of HSTL; however, they highlighted that some recent research has indicated that elderly individuals may experience HSTCL, differently from what has been previously described. Additionally, although it has little clinical significance, dysplastic bone marrow mimicking myelodysplastic neoplasms was described in marrow smears of HSTCL patients.

## 7. Anaplastic Large-Cell Lymphomas (ALCLs)

Anaplastic large-cell lymphoma (ALCL) ALK+, anaplastic large-cell lymphoma (ALCL) ALK−, and breast-implant-associated (BI) anaplastic large-cell lymphoma were all classed together in the ICC and the WHO-HAEM5.

Anaplastic large-cell lymphomas (ALCLs) are all characterized by large cells, often pleomorphic, and with horseshoe-shaped nuclei and copious cytoplasm, which are so-called hallmark cells. They typically express CD30 and frequently present with a deficient expression of T-lineage markers [51,52].

In anaplastic lymphoma kinase (ALK)+, the ALK protein is consistently expressed with variable cellular location, based on the different chromosomal rearrangements involving the *ALK* gene and different partner genes [53].

ALK-negative (ALK−) ALCL, which lacks the ALK protein and *ALK* rearrangement, represents a distinct subtype of anaplastic large-cell lymphoma. It was in fact well recognized that ALK-ALCL was a diverse entity, distinct from other PTCLs [54]. Several recurring genomic changes have been found in ALK-ALCL. The most frequent lesion found was *PRDM1* inactivation caused by chromosomal losses or somatic mutations. *PRDM1* encodes for one of the master regulators of T-cell differentiation, and appears to be a key gene in the pathogenesis of ALC [55]. On the other hand, STAT3 activation plays a major role, and is an effect of various genomic rearrangements [56]. Furthermore, *TP63* rearrangements have been found in ALK-negative ALCL, with similar effects to *TP53* loss, and negative prognostic impact.

A genetic subtype of systemic ALK-negative anaplastic large-cell lymphoma known as *DUSP22*-rearranged ALCL has been identified, carrying a relatively better prognosis [57]. The role of additional *JAK2* or *TP63* rearrangements in association with *DUSP22* rearrangements is under evaluation. *LEF1* expression seems to also be involved in the molecular pathogenesis of ALCL, and its neoplastic cells are characterized by a “doughnut cell” morphology [58] and a sheet-like development pattern with less pleomorphic cells [59].

ALCL linked to breast implants (BIA-ALCL) is more often a non-invasive neoplasm that develops in association with textured-surface breast implants and is associated with an excellent outcome; less frequently, the invasion of other adjacent structures is observed and is typically associated with a worse prognosis [60]. BI-ALCL has been upgraded from a provisional to a definite entity separated from the other two ALK-ALCLs in the two classifications. Numerous hypotheses attempted to explain the role of T-Helper2’s (TH2) allergic inflammatory response and immune evasion through amplification of 9p24.1 or overexpression of PDL1, which are observed in more than 50% of cases; however, none were conclusive. The molecular pathogenesis seems to be characterized by constitutive JAK-STAT activation, as documented by gene expression analyses [61] and confirmed by the occurrence of activating somatic mutations of *STAT3*, *STAT5B*, *JAK1*, and *JAK2*, and loss-of-function mutations of *SOCS1* and *SOCS3* [62,63,64,65].

## 8. Nodal T-Follicular Helper Cell Lymphomas

WHO HEAM4R already grouped mature T-cell lymphomas with an immunophenotype related to that of T-follicular helper (TFH) cells [66,67,68,69], including angioimmunoblastic T-cell lymphoma (AITL), follicular helper T-cell lymphoma (FTCL), and PTCL/NOS with a TFH phenotype showing a diffuse or T-zone pattern without follicular dendritic cell (FDC) proliferation [3,7]. They are all characterized by a similar immunophenotype and a series of recurring genetic as well as epigenetic abnormalities (see below) [70,71,72]. Interestingly, this may reflect a common, specific sensitivity to demethylating agents [73].

Both the ICC and the WHO-HEAM5 have recommended a new nomenclature for this distinct subtype of nodal T-follicular helper cell lymphomas that can be distinguished based on a few parameters, including cell morphology and tissue architecture, the tumor microenvironment, and follicular dendritic cell (FDC) distribution.

The relation to TFH cells is determined at immunophenotyping, by detecting the expression of two or (better) three TFH phenotypic markers in neoplastic cells. The ones that are most frequently employed are PD1, ICOS, CD10, BCL6, and CXCL13 [68,74,75,76].

A recent gene expression profile analysis indicated that the three entities, despite being related, maintain individual features (M. Etebari et al., in press).

### 8.1. Nodal TFH-Cell Lymphoma, Angioimmunoblastic Type (nTFHL-AI)

The development of arborizing post-capillary arteries that are consistent with high endothelial venules is the disease’s distinguishing feature. Medium-sized, atypical lymphocytes with clear cytoplasm are usually observed, embraced by a hyperplastic FDC meshwork.

Numerous studies demonstrate a link between nTFHL-AI and clonal hematopoiesis, with frequent *TET2* (present in 30–40% of patients) and/or *DNMT3A* mutations, detected in both neoplastic lymphocytes and bone marrow cells. Other common mutations involve TCR signaling, *IDH2*, and most commonly the *RHOA* gene, with an *RHOAG17V* hotspot mutation [74,76,77]. The high frequency of such lesion makes it possible to use them to support diagnosis in difficult cases [78].

### 8.2. Nodal TFH-Cell Lymphoma, Follicular Type (nTFHL-F)

Included in past among PTCL/NOS as a morphological variant, nTFHL-F is characterized by clusters of atypical lymphoid cells with pale cytoplasm surrounded by small lymphocytes of the mantle zone type, forming nodular structures that resemble germinal centers, and substantial expression of PD1 as well other TFH markers in the tumor cells. It is frequently associated with a t(5:9)(q33;q22) [79,80] translocation leading to *ITK/SYK* rearrangement [53].

### 8.3. Nodal TFH-Cell Lymphoma Not Otherwise Specified (NOS)

This tumor is constituted of a sheet-like proliferation of medium- to large-sized malignant cells. It may assume a typical T-zone pattern (neoplastic cells surrounding preserved germinal centers) without FDC hypertrophy. In short, any nodal PTCL not fulfilling the diagnostic definition of nTFHL-AI or nTFHL-F should be diagnosed as nTFHL-NOS. The genetic features and phenotype reflect those of nTFHL-AI.

## 9. Other Peripheral T-Cell and NK-Cell Lymphomas

### 9.1. Extranodal NK/T-Cell Lymphoma

The WHO-HEAM5 dropped the “nasal-type” from the preceding nomenclature in WHO4 (extranodal NK/T-cell lymphoma, nasal type) due to the disease’s prevalence in various extranodal locales. This tumor is characterized by vascular injury and damage, substantial necrosis, cytotoxic phenotype, and association with EBV. Although most cases appear to be true NK-cell neoplasms, in some instances a cytotoxic T-cell phenotype is recorded.

ICC maintained the previous name, despite the evidence of frequent non-nasal cases [3,4].

### 9.2. PTCL-NOS (Peripheral T-Cell Lymphoma NOS)

The designations of peripheral T-cell lymphoma NOS (PTCL-NOS) are still in use in WHO5. In nodal PTCL, NOS is frequently identified by an exclusionary diagnosis process. Two molecular subtypes, PTCL-BX21 and PTCL-GATA3, which resemble T-helper type 1 (Th1) and Th2 cells, respectively, have been identified [69]. According to clinical studies, the PTCL-GATA3 subgroup’s prognosis is worse, and its genetics are usually complex [81]; conversely, the PTCL-TBX21 subgroup is characterized by a relatively better prognosis, fewer copy number alterations, and more mutations in the genes that regulate DNA methylation [81].

A provisional immunohistochemistry-based technique evaluating TBX21, CXCR3, GATA3, and CCR4 is used to classify these subgroups [69]. According to WHO-HAEM5, PTCL-GATA3 has a uniform molecular genetic profile, while PTCL-TBX21 is heterogeneous and may include a subgroup with a cytotoxic gene expression program and aggressive behavior [69,81]. The genetic context, clinicopathological context, and prognostic implications of these possible biological variants of PTCL-NOS are currently not well enough understood to warrant a formal designation as a separate “subtype”.

### 9.3. NK/T-Cell Intravascular Lymphoma

Intravascular NK/T-cell lymphoma was previously defined among the extranodal NK/T-cell lymphomas (ENKTLs). In WHO5, this EBV+ tumor is currently classified as an aggressive NK-cell leukemia rather than an extranodal NK/T-cell lymphoma. Further evidence will support a better definition in the future. Clinically, the disease is particularly aggressive with common involvement of skin and CNS [82,83,84].

### 9.4. Nodal EBV+ T- and NK-Cell Lymphoma

Both the ICC and the WHO-HAEM5 treat nodal EBV-positive T- and NK-cell lymphomas as distinct diseases. They are considered aggressive diseases with a dismal prognosis.

Typically, lymphadenopathy, either with or without extranodal involvement, advanced disease, and B symptoms, is recorded. This lymphoma typically resembles diffuse large B-cell lymphoma morphologically and lacks the angioinvasion and coagulative necrosis found in ENKTL. It frequently exhibits a cytotoxic T-cell immunophenotype, in contrast to NK cells. The genetic environment is distinct from ENKTL, and *TET2* is the gene that is most frequently mutated [85,86,87].

### 9.5. EBV-Positive and T- and NK-Cell Proliferations and Lymphomas of Childhood

According to WHO-HAEM4, this group of diseases includes severe mosquito bite allergy, hydroa vacciniforme-like lymphoproliferative disease, chronic active EBV infection of T- and NK-cell type, systemic form, and systemic EBV-positive T-cell lymphoma of childhood. This nomenclature was revised by both WHO HEAM5 and ICC; the term hydroa vacciniforme-like lymphoproliferative disorder has been modified to hydroa vacciniforme lymphoproliferative disorder (HVLPD). Additionally, systemic chronic active EBV disease (CAEBV) has substituted T- and NK-cell type systemic chronic active EBV infection. Severe mosquito bite allergy and childhood systemic EBV-positive T-cell lymphoma maintained their previous names.

These diseases are extremely rare in Europe and USA, with most reports being from Asia and Central America [88]. HVLPD presents as skin lesions in sun-exposed areas characterized by the presence of EBV-infected T or NK cells and exceptionally high levels of EBV DNA in the blood [15,89,90]. Caucasians are more likely to have classic forms of HVLPD, usually limited to the skin. Systemic symptoms (i.e., associated CABVD), conversely, are typically seen in Asian [91] as well as Hispanic populations [92,93,94].

CAEBV disease is defined by a persistent (more than three months) and worsening illness, elevated blood levels of EBV DNA, and organ infiltration by EBV-infected cells (histologically documented) without any apparent immunodeficiency [95,96,97]. Due to the highly aggressive course, often fatal if stem cell transplantation is not performed, the term “CAEBV infection” has been replaced with “CAEBV disease”.

Currently, the term “CAEBV” is only applied to describe NK-cell disease. In fact, B-cell CAEBV was found to be related to immunodeficiencies. According to multiple recent genomic studies, there is a likelihood that the CAEBV disease shares somatic mutations (e.g., *DDX3X* and *KMT2D*) also observed in T- and NK-cell lymphomas. Furthermore, the presence of intragenic deletions in the EBV genome raises the possibility that these modifications play a significant role in EBV-associated neoplasia [95,97,98,99,100].

## 10. Conclusions

The novel classification schemes presented in 2022 offered some significant changes to the definition of mature T-cell neoplasms. Unfortunately, a few discrepancies do exist (Table 2), and diagnostic reports should therefore specify the classification to which they are referring to. Indeed, as tumor classification is the basis of precision medicine, it is desirable that the next classification overcomes these differences as a unique language is mandatory for proper dialogue and scientific improvement.

## Figures and Tables

**Table 1 ijms-24-14170-t001:** Comparison of WHO-HEM5, ICC, and WHO-HEM4.

WHO-HEAM5	WHO-HEAM4R	ICC
**Mature T-cell and NK-cell neoplasms**		
**Mature T-cell and NK-cell leukemia**		
T-prolymphocytic leukemia	T-prolymphocytic leukemia	T-cell prolymphocytic leukemia
T-large granular lymphocytic leukemia	T-cell large granular lymphocytic leukemia	T-cell large granular lymphocytic leukemia
NK large granular lymphocytic leukemia	Chronic lymphoproliferative disorder of NK cells	Chronic lymphoproliferative disorder of NK cells
Adult T-cell leukemia/lymphoma	Adult T-cell leukemia/lymphoma	Adult T-cell leukemia/lymphoma
Sezary syndrome	Sezary syndrome	Sezary syndrome
Aggressive NK-cell leukaemia	Aggressive NK-cell leukaemia	Aggressive NK-cell leukemia
**Primary cutaneous T-cell lymphomas**	**Primary cutaneous T-cell lymphomas**	
Primary cutaneous CD4-positive small or medium T-cell lymphoproliferative disorder	Primary cutaneous CD4-positive small or medium T-cell lymphoproliferative disorder	Primary cutaneous small/medium CD4 T-cell lymphoproliferative disorder
Primary cutaneous acral CD8-positive lymphoproliferative disorder	Primary cutaneous acral CD8-positive T-cell lymphoma	Primary cutaneous acral CD8 T-cell lymphoproliferative disorder
Mycosis fungoides	Mycosis fungoides	Mycosis fungoides
Primary cutaneous CD30-positive T-cell lymphoproliferative disorder: lymphomatoid papulosis	Primary cutaneous CD30-positive T-cell lymphoproliferative disorder: lymphomatoid papulosis	Primary cutaneous CD30-positive T-cell lymphoproliferative disorder: lymphomatoid papulosis
Primary cutaneous CD30-positive T-cell lymphoproliferative disorder: primary cutaneous anaplastic large-cell lymphoma	Primary cutaneous CD30-positive T-cell lymphoproliferative disorder: primary cutaneous anaplastic large-cell lymphoma	Primary cutaneous anaplastic large-cell lymphoma
Subcutaneous panniculitis-like T-cell lymphoma	Subcutaneous panniculitis-like T-cell lymphoma	Subcutaneous panniculitis-like T-cell lymphoma
Primary cutaneous gamma/delta T-cell lymphoma	Primary cutaneous gamma/delta T-cell lymphoma	Primary cutaneous gamma-delta T-cell lymphoma
Primary cutaneous CD8-positive aggressive epidermotropic cytotoxic T-cell lymphoma	Primary cutaneous CD8-positive aggressive epidermotropic cytotoxic T-cell lymphoma	Primary cutaneous CD8 aggressive epidermotropic cytotoxic T-cell lymphoma
Primary cutaneous peripheral T-cell lymphoma, NOS	Not previously included	
**Intestinal T-cell and NK-cell lymphoid proliferations and lymphomas**		
Indolent T-cell lymphoma of the gastrointestinal tract	Indolent T-cell lymphoproliferative disorder of the gastrointestinal tract	Indolent T-cell lymphoma of the gastrointestinal tract
Indolent NK-cell lymphoproliferative disorder of the gastrointestinal tract	Not previously included	Indolent clonal T-cell lymphoproliferative disorder of the gastrointestinal tract
Enteropathy-associated T-cell lymphoma	Enteropathy-associated T-cell lymphoma	Enteropathy-associated T-cell lymphoma Type II refractory celiac disease
Monomorphic epitheliotropic intestinal T-cell lymphoma	Monomorphic epitheliotropic intestinal T-cell lymphoma	Monomorphic epitheliotropic intestinal T-cell lymphoma
Intestinal T-cell lymphoma, NOS	Intestinal T-cell lymphoma, NOS	Intestinal T-cell lymphoma, NOS
**Hepatosplenic T-cell lymphoma**		
Hepatosplenic T-cell lymphoma	Hepatosplenic T-cell lymphoma	Hepatosplenic T-cell lymphoma
**Anaplastic large cell lymphomas**		
ALK-positive anaplastic large-cell lymphoma	Anaplastic large-cell lymphoma, ALK-positive	Anaplastic large-cell lymphoma, ALK positive
ALK-negative anaplastic large-cell lymphoma	Anaplastic large-cell lymphoma, ALK-negative	Anaplastic large-cell lymphoma, ALK negative
Breast-implant-associated anaplastic large-cell lymphoma	Breast-implant-associated anaplastic large-cell lymphoma	Breast-implant-associated anaplastic large-cell lymphoma
**Nodal T-follicular helper (TFH)-cell lymphoma**		Follicular helper T-cell lymphoma
Nodal TFH-cell lymphoma, angioimmunoblastic-type	Angioimmunoblastic T-cell lymphoma	TFH lymphoma, angioimmunoblastic type (angioimmunoblastic T-cell lymphoma)
Nodal TFH cell lymphoma, follicular-type	Follicular T-cell lymphoma	Follicular helper T-cell lymphoma, follicular type
Nodal TFH cell lymphoma, NOS	Nodal peripheral T-cell lymphoma with TFH phenotype	Follicular helper T-cell lymphoma, NOS
**Other peripheral T-cell lymphomas**		
Peripheral T-cell lymphoma, not otherwise specified	Peripheral T-cell lymphoma, not otherwise specified	Peripheral T-cell lymphoma, NOS
**EBV-positive NK-/T-cell lymphomas**		
EBV-positive nodal T- and NK-cell lymphoma	Not previously included	
Extranodal NK-/T-cell lymphoma	Extranodal NK-/T-cell lymphoma, nasal-type	
**EBV-positive T- and NK-cell lymphoid proliferations and lymphomas of childhood**		
Severe mosquito bite allergy	Severe mosquito bite allergy	Severe mosquito bite allergy
Hydroa vacciniforme lymphoproliferative disorder	Hydroa vacciniforme-like lymphoproliferative disorder	Hydroa vacciniforme lymphoproliferative disorder Classic Systemic
Systemic chronic active EBV disease	Chronic active EBV infection of T- and NK-cell type, systemic form	Chronic active Epstein–Barr virus disease, systemic (T-cell and NK-cell phenotype)
Systemic EBV-positive T-cell lymphoma of childhood	Systemic EBV-positive T-cell lymphoma of childhood	Systemic Epstein–Barr-virus-positive T-cell lymphoma of childhood

**Table 2 ijms-24-14170-t002:** Highlights of the main differences between the two classifications.

1	The WHO-HAEM5 groups mature T-cell malignancies into nine categories, while the ICC puts all entities into one group of mature T-cell and NK-cell neoplasms.
2	The WHO-HAEM5 changed the nomenclature of T-cell large granular lymphocytic leukemia, chronic lymphoproliferative disorder of NK cells, anaplastic large-cell lymphoma, and ALK-positive and -negative anaplastic large-cell lymphoma, but ICC retained the old nomenclature from the WHO-HAEM4R.
3	The WHO-HAEM5 included primary cutaneous peripheral T-cell lymphoma NOS and EBV-positive nodal T- and NK-cell lymphoma as new entities, while the ICC did not include them.
4	Regarding intestinal T-cell and NK-cell lymphoid proliferations and lymphomas, both the WHO-HAEM5 and the ICC revised the name of indolent T-cell lymphoproliferative disorder of the gastrointestinal tract, now included as indolent T-cell lymphoma of the gastrointestinal tract, and both classifications added indolent NK-cell lymphoproliferative disorder as a new entity.
5	WHO-HAEM5 groups primary cutaneous T-cell lymphoid proliferations and lymphomas (CTCL) into nine different entities, while in the ICC, the sole observed change was to the nomenclature of primary cutaneous acral CD8+ T-cell lymphoma, as it is now classified as a primary cutaneous acral CD8+ T-cell LPD.

## Data Availability

Not applicable.

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
