# Peer review of "A Comparison of the Fifth World Health Organization and the International Consensus Classifications of Mature T-Cell Lymphomas"

_ijms, 2023, doi:10.3390/ijms241814170_

Round 1
Reviewer 1 Report
The paper by Piccaluga and Khattab compares two recent classifications of mature T-cell lymphomas elaborated by the 5th World Health Organization and the International Consensus classifications of mature T-cell lymphomas. Although the classifications are quite similar, there are some differences which are worth to be highlighted, and should be kept in mind when performing research and treating patients. Hopefully, in the future a common classification will be prepared, which facilitates comparison of the results of research and clinical studies.
It has not been indicated that the WHO-HAEM5 groups T-cell malignancies into: mature T-cell and NK-cell leukaemias, primary cutaneous T-cell lymphomas, intestinal T-cell and NK-cell lymphoid proliferations and lymphomas, nodal T-follicular helper (TFH) cell lymphoma, etc; while the ICC puts all entities into one group of mature T-cell and NK-cell neoplasms.
In WHO-HAEM5 classification mycosis fungoides and Sézary syndrome are put into different groups, while in the ICC they are listed alongside.
A table should be included to highlight the differences among the three classifications; WHO-HAEM4, WHO-HAEM5 and ICC.
43 Citations 2 and 5 do not include classification of T-call malignancies, and should be deleted
77 “founded in T_PLL” should read “found in (?)% of T-PLL”
The are several linguistic errors that should be corrected.
Author Response
We are grateful to all revieweres and the Editor for the useful advices. We modified the text according to all of them. Please find our point by point reply in the following:
- It has not been indicated that the WHO-HAEM5 groups T-cell malignancies into: mature T-cell and NK-cell leukaemias, primary cutaneous T-cell lymphomas, intestinal T-cell and NK-cell lymphoid proliferations and lymphomas, nodal T-follicular helper (TFH) cell lymphoma, etc; while the ICC puts all entities into one group of mature T-cell and NK-cell neoplasms.
Response
These Paragraphs has been added to the introduction section and the different approach is visible in Table (1)
- In WHO-HAEM5 classification mycosis fungoides and Sézary syndrome are put into different groups, while in the ICC they are listed alongside.
Response
This is correct. We indicated this difference in the text and within the Table 1.
- A table should be included to highlight the differences among the three classifications: WHO-HAEM4, WHO-HAEM5 and ICC.
Response
A table has been added for comparison (lane 59). We apologize as the Table was already made but not visible in the first submission.
- Citations 2 and 5 do not include classification of T-call malignancies, and should be deleted
Response
We included them as part of the entire classification of hematological malignancies, that include also myeloid disorders.
- “founded in T_PLL” should read “found in (?)% of T-PLL
Response
we corrected it and updated the reference.
- The are several linguistic errors that should be corrected
Response
We apologize. Typos were corrected
Reviewer 2 Report
This is an excellent review and comparison between the current new classifications, that will aid the readers to understanding of differences and the pathologocial and clinical implications hereoff.
Author Response
We deeply thank the reviewer for the positive evaluation
Reviewer 3 Report
Piccaluga and Khattab have reviewed extensively a comparison of the 5th World Health Organization and the International Consensus classifications of mature T-cell lymphomas. It is a very informative review, and it emphasizes a common nomenclature to unify the two classifications is mandatory to ensure precise medicine. It is recommended that the authors address the following questions.
Major issues:
1) This review should have a separate category discussing nodal T-cell lymphoma. “4.5. Nodal T-follicular helper cell lymphomas” should not belong to “4. Extranodal non-leukemic mature T-cell neoplasm” because it is nodal T-cell lymphoma.
2) The review should have a table or tables comparing the differences between WHO-5 and ICC 2022.
Minor issues:
There are multiple typos and format errors, for example:
1) Page 67 mentions Table 1. I did not see Table 1 in the manuscript.
3) Page 77, “T_PLL” should be “T-PLL”.
4) Page 77 “T-cell phenotype that is negative for”. “that” should be deleted.
5) “the neoplastic cells of T-LGL” should be “the neoplastic cells of T-LGL leukemia”
6) Page 123, there should be a period between “Romania” and “It’s”
7) Page 134, “neoplams” should be “neoplasm”
There are some typos and formatting errors, which need to be corrected.
Author Response
We are grateful to all revieweres and the Editor for the useful advices. We modified the text according to all of them. Please find our point by point reply in the following:
- This review should have a separate category discussing nodal T-cell lymphoma. “4.5. Nodal T-follicular helper cell lymphomas” should not belong to “4. Extranodal non-leukemic mature T-cell neoplasm” because it is nodal T-cell lymphoma.
Response
we fixed the paragraphs numbering and we put it in a separate category.
- The review should have a table or tables comparing the differences between WHO-5 and ICC 2022.
- Page 67 mentions Table 1. I did not see Table 1 in the manuscript.
Response
A table has been added for comparison (lane 59). We apologize as the Table was already made but not visible in the first submission.
- Page 77, “T_PLL” should be “T-PLL”. Corrected
- Page 77 “T-cell phenotype that is negative for”. “that” should be deleted. Deleted
- “the neoplastic cells of T-LGL” should be “the neoplastic cells of T-LGL leukemia”. Corrected
- Page 123, there should be a period between “Romania” and “It’s”. Corrected
Reviewer 4 Report
Dear Authors,
Well, I am not a fan about "comparison" manuscripts, but the topic is really significant. However, I would like the authors to underline the novelty of the manuscript more because it is not understandable at all. I also missed the general picture about that how often the existing classifications are used and in which part of the world. I would like to ask the authors to give some insight about this in one two paragraphs of Introduction.
Additionally, please add the following: 1) Introduction. There is an aim, but refer please also the provided methodology, - key words, time duration for the work, inclusion/exclusion criteria for the literature sources.
2) Conclusions. I would like to ask the authors also to develop more precise Conclusions. You mentioned "a few discrepancies", - please, shortly give info about them also in the Conclusions.
3) Finally, this manuscript is more of the descriptive character than scientific thus, I guess it should be submitted is another more suitable Journal for such manuscripts.
Author Response
We are grateful to all revieweres and the Editor for the useful advices. We modified the text according to all of them. Please find our point by point reply in the following:
- I would like the authors to underline the novelty of the manuscript more because it is not understandable at all.
- I also missed the general picture about that how often the existing classifications are used and in which part of the world. I would like to ask the authors to give some insight about this in one two paragraphs of Introduction.
Response
This is the first article comparing the two classifications as far as TCL are concerned. This is therefore the first practical guide to pathologists in this regard. We specified it in the introduction
There are no official info about how the two classifications are currently applied worldwide. We specified in the texts and recommended such analysis
- There is an aim, but refer please also the provided methodology, - key words, time duration for the work, inclusion/exclusion criteria for the literature sources.
Response
We added some pieces of information; please note, however, that we basically compared two manuscripts, other literature is not matter of systematic analysis. Only refers to additional information about the entities.
- I would like to ask the authors also to develop more precise Conclusions. You mentioned "a few discrepancies", - please, shortly give info about them also in the Conclusions.
Response
We added a table with detailed comments
- Finally, this manuscript is more of the descriptive character than scientific thus, I guess it should be submitted is another more suitable Journal for such manuscripts
Response
We respect the reviewer’s point of view. However, the subject was discussed with the guest editor before submission.
Round 2
Reviewer 4 Report
Dear Authors,
Well, thanks, I really appreciate the changes and especially Conclusions! Excellent!
I will advice to publish this manuscript if it really fits within the frame of Journal.